# Impact of workplace bullying on work engagement among early career employees

Iqra Javed[1], Amna Niazi [2]*, Sadia Nawaz[3], Muhammad Ali[4], Mujahid Hussain[4]

1 Institute of Business & Management, University of Engineering & Technology, Lahore, Pakistan,
2 Humanities and Social Sciences Department, University of Engineering & Technology, Lahore, Pakistan,
3 Department of Human Centred Computing, Faculty of IT, Monash University, Melbourne, Australia,
4 FAST School of Management, National University of Computer & Emerging Sciences, Lahore, Pakistan

* amna.niazi@uet.edu.pk

**Data Availability Statement:** Data cannot be shared publicly because of confidentiality. However, it can be available upon request. Data contain potentially identifying material through which the employee may be identified. If the data

## Abstract

This study aims to measure the impact of workplace bullying on work engagement in terms of employee silence and knowledge sharing. It also helps to explain how psychological contract breach moderates the bullying-silence relationship. For this study, data is collected from 384 early-career employees having experience up to three years from seven banks of Lahore, Pakistan. Findings of this study reveals that workplace bullying has a positive relationship with employee silence and negative relationship with work engagement. Results of all moderation and mediated variables are significantly related to each other. However, the results explain that a psychological contract breach slightly moderates the bullying-silence relationship. Survey-based questionnaire, cross-sectional research design, and convenience-based sampling technique are some of the limitations of this study. This is the first study that tried to investigate the bullying-engagement relationship among early-career employees in the banking sector of Lahore, Pakistan. This study may help practitioners and policymakers to develop anti-bullying laws that can support the management in overcoming the negative workplace environment. This study aims to promote an equal opportunity for all employees where they can raise their voices about misconduct. This is the first study that investigated the victimization of bullying behavior among early-career employees in a Pakistani cultural context.

## Introduction

Workplace bullying has emerged as a very strong employment concern. Employee's well-being and productivity is greatly hampered by bullying at workplace. Organizations today are striving to create a bullying free workplace, where employees are encouraged to express their individuality and creativity without such nuisances. Workplace bullying is a psychological, social and organizational issue confronted by employees at the workplace. Workplace bullying is defined as the repetitive actions that are pointed towards someone, which is a cause for embarrassment for that person. Sometimes workplace bullying is done intentionally and sometimes unintentionally. However, it is a major cause for stress for that person which effects the

needs to be accessed than that can be accessed by sending an email to ghulam.muhammad70@nu. edu.pk (the secretary of the ethics committee of FAST).

**Funding:** The author(s) received no specific funding for this work.

**Competing interests:** The authors have declared that no competing interests exist.

personal performance and employees mental health [1, 2]. It is observed that employees will perform better, if they are psychologically and mentally fit [3]. According to a research report, world victimization of bullied employees has increased from 75% in 2008 to 94% in 2019, indicating an upward trend in workplace bullying [4]. It was found that employees were bullied through different sources such as hostile email tones, colleagues' negative remarks and the shouting of managers at employees. Excessive workload, negative remarks and unrealistic deadlines are other common forms of workplace bullying [5]. Women were reported to be bullied more frequently and suffer greater damages. However, only 17% of cases were delt by an internal inquiry committee of the organization [6].

Bullying destroys the mental peace, emotional orientation and creates unrest among employees especially in the service sector where there is a close connection between the employees and the customers. The disturbed employees then lose their focus and productivity and are unable to satisfy their customers' needs [7]. Bullying is negatively related to performance, commitment, and task involvement of employees [8]. When employees are mistreated, they show less motivation, are involved in counter-productive workplace behavior and less likely to engage in productive activities [9]. Research also reveals that workplace bullying causes adverse consequences on employee's mental, physical, and emotional health [10, 11]. Since, past scholars highlighted the bullying-engagement relationship in terms of commitment, performance and customer satisfaction. It is interesting to find out how bullied employees are going to behave when it comes to knowledge sharing within the organization. Some employees choose to remain silent while others choose to express their displeasure by communicating their reservations. The reaction of employee who is bullyied may vary but this has short term and long term consequences. Literature is available that explains the workplace bullying has a negative impact on the performance of the employee however, its relationship with knowledge sharing and employee silence is not studied earlier especially when the psychological contract breach impacts the relationship as a moderator.

The present study uses conservation of resource (COR) theory to examine the indirect relationship of employee silence and knowledge sharing among bullying-engagement relationships [12]. COR theory illustrates that early-career employees try to conserve their physical (information and knowledge) and psychological (emotional stability and mental health) resources in anticipation of future damage. When employees are mistreated, some of them adopt a passive-coping strategy by being silent to avoid stress and loss. Such employees are less likely to be effectively engaged in productive activities [9]. These employees feel that their manager has abused his power to victimize them and has breached the psychological contract (PCB). In literature, PCB is explained as a contract between two parties' (employer and employee) in which early-career employees perceive that their manager is not able to fulfill their demands that were promised [13]. This concept is based on the social contract theory. When the manager breaches the contract by not providing justice or a supportive environment then as a reaction early-career employees show silent behavior and less attention towards their day-to-day task accomplishment [14]. The current study takes psychological contract breach as a moderator between workplace bullying and employee silence. Psychological contract breach refers to a violation of the implicit or explicit agreement between an employee and their employer regarding the terms of their employment. When the psychological contract is breached, employees may feel disillusioned, frustrated, or betrayed, and these feelings can contribute to a variety of negative outcomes, including decreased employee engagement, increased employee silence, and reduced knowledge sharing which can in turn perpetuate the cycle of abuse. Additionally, when employees feel that the psychological contract has been breached, they may be less likely to trust their employer or coworkers, and this can reduce the willingness of employees to share knowledge or collaborate with one another. By understanding the role

of psychological contract breach in the relationship between workplace bullying, employee silence, knowledge sharing, and employee engagement, organizations can work to create a more positive workplace environment and improve the overall wellbeing of their employees.

The aim of the present study is to measure the effect of workplace bullying among early-career employees with the mediating role of employee silence and knowledge sharing and its effect on their work engagement in the banking sector, Lahore-Pakistan. However, the moderating effect of psychological contract breach is also measured among bullying-silence relationship. Contextual factors like tenue, sector and experience of the employees were also observed since they help to provide a more complete understanding of the underlying relationships and potential biases that may exist. The tenure of the employees with the company was observed since this research focused on early career employees. Employees who have been with a company for a longer period may have different perspectives and experiences compared to those who have just started, and this can impact the relationship between variables. Banking industry was focused since it is one of the main service sectors of Pakistan to run the economy and provide employment opportunities for young employees. To test the proposed model, hypotheses are proposed and later tested with empirical data.

The present study contributes to the workplace bullying literature in multiple ways. Previous research has solely focused on workplace bullying in service sector employees [7]. The current study highlights the victimization of bullying behavior among early career-employees in the banking sector of Pakistan. This industry was selected since the banking industry plays an important role in the economic system of any country. The context of the Pakistani banking sector presents a unique set of challenges and opportunities for studying the relationship between workplace bullying, employee silence, knowledge sharing, and employee engagement. Compared to previously studied contexts, the comparability with the Pakistani banking sector is severely limited due to several contextual factors that are specific to this region. The Pakistani banking sector operates within a different cultural, political, and economic environment, and these factors can influence the prevalence and nature of workplace bullying, as well as the attitudes and behaviors of employees in response to bullying. The present research can help Pakistani banking sector to develop more effective strategies for addressing workplace bullying and improving employee engagement, and it can also inform the development of policies and regulations that can help to reduce the incidence of workplace bullying in this sector. Additionally, by examining the relationship between these variables in a new context, we can broaden our understanding of these relationships more generally, and this can inform future research and help to build a more comprehensive understanding of the factors that influence workplace bullying and employee engagement.

Early career employees were selected since it is observed that employees who are experienced are able to handle the bullying situations in a better way and hence, early career employees are the ones who are naïve and are not able to cope up with such situations. Employers try to control bullying at the workplace, but it is difficult and thus continues to go unnoticed. Another contribution is the detailed discussion on the mediating role of employee silence among workplace bullying and work engagement which has been a topic that has received considerable scholarly attention [15–17]. Engagement is defined as a mindset that is satisfying and dedicated to achieving a task. Its major components include vigor, dedication and absorption [18]. This study views workplace bullying as two-dimensional construct, i.e., work-related bullying and person-related bullying [19]. Work-related bullying consists of a behavior which may include setting unrealistic deadlines for the employee or assigning an employee a task which is below the competency level of the employee. Person-related bullying includes a behavior which may include excluding an employee from the group, intentionally spreading false information about the person and insulting the person by making fun [20].

The study is unique in presenting the idea of workplace bullying and its impact on employee engagement for several reasons. Firstly, the study provides insight into the impact of workplace bullying on early-career employees in the banking sector in Lahore, Pakistan, which has not been well researched previously. Secondly, the examination of employee silence as a mediator between bullying and work engagement adds to the understanding of the psychological processes underlying the effects of bullying on employees. Additionally, the examination of knowledge sharing as an effect of bullying and employee silence highlights the potential negative impact of bullying on organizations through decreased knowledge sharing among employees. Finally, the study's focus on the moderating role of psychological contract breach in the relationship between bullying, employee silence, and work engagement is innovative and sheds light on the importance of considering the broader organizational context in understanding the effects of bullying.

## Workplace bullying and employee silence

The word employee silence is explained as intention concealment of suggestions, ideas, or organization- related information that is beneficial for organizational productivity [21]. Past literature has explained employee silence in terms of different dimensions i.e., defensive silence, acquiescent silence, and ineffectual silence. However, defensive silence is defined as when targeted employee is reluctant to share their opinion due to adverse feedback from managers [22]. A study was conducted in Korean hotels to measure the impact of acquiescent silence on bullying among managers and employees. Results confirmed that contractual workers in hotels have experienced less intimidation and negative rumors as compared to permanently hired employees [23]. Furthermore, the authors have tried to investigate the bullying-silence relationship in the Indian cultural context with different variables like workplace friendship and psychological contract violation in the banking sector [9]. Results of this study concluded that friendly behavior in banks creates an indirect and less negative effect on bullying-silent relationship. As the supportive working climate motivates banking employees to raise their voice about any misconduct [9].

Further, conservative resource theory (COR) helps to explain the fact that why employees try to conserve their resources in the form of hiding information [24]. Resource depletion is minimized when organizations have created an autonomous culture and supportive working climate. Furthermore, past research on the bullying-silence relationship shows that personality plays a vital role in the determination of abusive behaviors i.e., employees who are calm and cool in their personality traits show less aggressive behavior and observed silence. They do not react violently, even if they face uncertain conditions [25]. Building on these arguments, it is deduced that in the service sector especially banks collectivist culture and personality of employee are the main determinants that directly affect the employees' silence behavior. In addition, a past study that has investigated the mediating effect of defensive silence among workplace ostracism and interpersonal deviance depicts that the self-protective behavior of employees indirectly affects interpersonal deviance and silent behavior among employees [26]. Researchers also highlighted that some employees are not willing to keep quiet for reporting of any workplace misconduct. They feel that negative activities should be highlighted in front of management so, as to lessen the adverse consequences in the future [27]. Studies were conducted in the past to measure the effect of bullying behavior in government banks among minorities in Asian countries [28]. Literature reveals that employees who are part of the minority groups are more mistreated and verbally abused than employees who belong to majority groups. This is one of the major reasons why they behave differently and react silently at their workplace [29]. While, if the organization have fair and equitable climate, everyone has

autonomy and power to speak about their problems then employees gain confidence and exhibit more commitment [30]. Thus, the following relationship is hypothesized from the above discussion.

H$_1$: There is a positive impact of workplace bullying on employee silence.

## Employee silence and work engagement

Work engagement has been defined in the literature, as a positive and satisfactory job-related attitude that guides affirmative actions. It is considered as absorption, vigor, and dedication towards work or assigned task [31]. Past research suggests that engaged employees have shown more commitment, high productivity, and lead towards fewer buyouts in organizations [22]. Research was conducted in the banking sector of India, to measure the impact of silent behavior of employees on their work engagement [32]. This study suggested that in private sector banks staff members are overburdened, unfair evaluation system prevail, and there is no channel to raise employees voice [33]. This had an adverse effect on employee's attitude towards their task accomplishment, de-motivated them and eventually led them to engage in self-protective behavior [32]. Hence, based on different arguments, it is inferred that in the banking sector employees mostly adopt ineffectual silence behavior due to an unsupportive work climate which ultimately reduces the internal motivation to perform the task with dedication [34]. It also affects the employee-client relationship since, if employees would feel secure and protected then customer service in the form of work engagement would increase. Self-determination theory states that empowered employees help to break down the silence, boost morale and provide self-motivation for task completion [35]. Satisfied employees perform their duties with more courage and dedication. Therefore, effective leadership has a direct impact on employees' engagement because productive leadership (authentic, charismatic, and transformational) can help to increase job satisfaction and work engagement [36]. Internally satisfied employees are self-motivated and highly engaged towards their task accomplishment. Hence, self-motivation is the key factor for the enhancement of work engagement [37]. Further, employee's engagement is directly affected by organizational culture. If the culture is supportive where everyone can openly share their ideas and report their problem without any fear, then organizational productivity could be enhanced [38]. Past research suggests that organizational culture varies among countries and societies because in high power distance culture employees are not involved in decision making and have no job autonomy [39]. This negative attitude has increased counterproductive work behavior. Due to this, employees engage in activities like treating customers offensively, spreading bad word of mouth about banking products, etc. [33]. However, due to the counterproductive behavior of supervisor, mostly employees are frightened to raise their voice about issues and problems, which negatively effects the employees' health in form of depression, anxiety, sleep disorder, poor customer dealing, and lack of interest in organizational activities [37]. Research has shown that there is a strong negative relationship between employee silence and work engagement. In other words, when employees are silent, they are less likely to be fully engaged in their work. This is because when employees feel that their voices are not being heard, they may feel disconnected from their work and the organization. They may become disengaged, lose motivation, and ultimately become less productive [40]. Further, to support the above relationship, a past study has elaborated that sometimes employees adopt passive behavior of silence in which they endeavor their relationship with managers. Pro-social silencing behavior is not based on self-interest [41]. But it negatively affects the level of engagement among employees. Therefore, after reviewing different arguments, it is depicted that in the service sector especially in banks

either public or private, managers should create a supportive and encouraging workplace environment. So, that early-career employees are less affected and show more courage and dedication in their task accomplishment and customer dealings. Hence, the following hypotheses are inferred from the above discussion.

H₂: There is a direct relationship between employee silence and work engagement.

H₃: Employee silence mediates the relationship between workplace bullying and work engagement.

## Workplace bullying and knowledge sharing

Bullying behavior in an organization has a major impact on the overall productivity and effectiveness of employees [42]. Recent research concludes that employees who experience a supportive working environment and peer coordination are encouraged to share beneficial information and ideas [43]. According to the knowledge-based view of organizational learning, the competitiveness of any firm can be judged by the quality of shared information and fairness in communication [44]. Another research conducted in high technological firms in Korea, measured the relation between bullying practices and information-sharing behavior among employees. Results suggested that victimized employees engage in negative work-related behaviors and hide information [45]. Therefore, it is argued that in the service sector, an adverse workplace environment in any form negatively affects the knowledge-sharing activities in the organization. To support this bullying-knowledge-sharing relationship different supporting theories were inter-connected. According to social cognitive theory (SCT), human is a social animal and cannot acquire knowledge alone without any external interference [46]. Due to rapid technological and structural changes, employees are dependent on each other for resources and information sharing. Therefore, in the case of the banking sector when employees are exposed to bullying behavior by their colleagues and managers, they tend to hide information. Not only hiding information but some of the other consequences include job burnout, turnover intensions, disloyalty and deviant work-related behavior [47]. According to social exchange theory, employees adjust their relationship with supervisors and colleagues based on self-interest i.e., cost-benefit relation [48]. For example, if employees are threatened and abused by their supervisor or peers then as a consequence victimized employees would not share information [49]. Previous research has also investigated the relation of knowledge sharing practices, psychosocial hazards, and organizational environment in the banking sector [50]. Results revealed that staffs in banks are overburdened, with excessive humiliation and negative feedback from managers. This attitude adversely affects knowledge sharing behavior within organization [26]. Keeping in view these arguments, it is revealed that in banking sector where power distance is high, employees are mistreated and disgraced at each stage which negatively affects the effective information sharing culture. It also disturbs the learning and innovative culture in organization. Form the above arguments, following hypothesis is derived.

H₄: Workplace bullying has a direct relationship with knowledge sharing.

## Knowledge sharing and work engagement

According to Hendriks *et al*., (2016), knowledge management includes conception, accessibility, and transfer of information from one level to another. These practices have been considered as very effective in knowledge distribution among employees [51]. This also motivates them to do their work effectively and efficiently. According to the social exchange theory, shared information provides mutual benefits for both the parties. When information is shared one individual shares productive information and the transferor in return receives an ultimate

reward in the form of performance improvement, bonuses and performance appraisal [48]. Research conducted in the IT sector of India also endorsed that knowledge sharing plays a significant mediating role between work engagement and organizational justice [23]. Empirical results concluded that information-sharing mechanisms have directly mediated the relationship between these two constructs. Highly committed and engaged employees are more creative and energetic as compared to disengaged employees [23]. Knowledge sharing is considered like a bridge that creates a positive effect in many forms and enhances employees' performance, motivation, loyalty, and organizational citizenship behavior [52]. Based on the above-mentioned arguments it is depicted that knowledge sharing behavior among bankers has enhanced the performance of the employees and work engagement also significantly plays a mediating role between them. Knowledge sharing can have a positive impact on work engagement by promoting a sense of belonging, enhancing learning and development, improving collaboration and teamwork, and increasing innovation and creativity. By fostering a culture of knowledge sharing, organizations can create a more engaged and productive workforce. Therefore, the following hypotheses are inferred from the above argument.

$H_5$: There is a direct relationship between knowledge sharing and work engagement.

$H_6$: Knowledge sharing mediates the relationship between workplace bullying and work engagement.

## Psychological contract breach as a moderator

Literature defines PCB as a state in which employees perceive that organizations have not met their promises or obligations [53]. A contract breach may occur when the organization is not able to provide a committed bonus, promotion, better working environment, etc. A study was conducted in Malaysian banking sector to check the relationship between psychological contract breach and organizational commitment. The results revealed that when the employee promises are fulfilled, they showed more commitment, engagement in-role behaviors and positive work-related outcome was achieved. On the other hand, if contract is breached then employees would engage in counterproductive work behaviors and exhibit low performance [38]. There are chances that they choose silent behavior due to the non-approval of information by authority. The effect of bullying on employee silence behavior varies depending on the level of breaching of the psychological contract. If the breach is low, bullying may have a more significant effect on employee silence behavior because the employee may have had positive expectations about their employer that are now being shattered by the bullying. In contrast, if the breach is high, the employee may already be indulged in silent behavior due to the employer's failure to meet their promises, and the bullying may simply exacerbate this. In either case, both breaching of a psychological contract and workplace bullying can lead to an employee exhibiting silent behavior, which can have significant consequences for the individual employee, as well as the organization as a whole. This concept is embedded in the social exchange theory and it is believed that, the psychological contract is a two-way process in which employer fulfills their promises (in either monetary or behavioral terms) and in return employees perform their obligations more satisfactorily [54]. Moreover, in the conservation of resource theory (COR), the author argued that higher job demands would lead to greater contractual violations by the supervisors [12]. This theory supported the literature of psychological contract breach in the Pakistani banking sector and proved that employees would be more engaged in silent behavior due to loss of organizational commitment [55]. Hence, based on the above arguments it is conferred that in the service sector employee's psychological contract is based on a mutual cost-benefit relationship. It shows that if the manager does not improve

working conditions the staff would not show commitment, enthusiasm, and most likely indulge in silent behavior.

A study conducted by Bari *et al.* (2020) in the service sector to measure the mediating effect of psychological contract breach in between employee silence and knowledge hiding resulted that knowledge hiding behavior was the major reason for employee silence and contract breach because in this situation employees were de-motivated, exhausted and eventually tend to quit the job [56]. Hence, based on these arguments it is inferred that when employees' implicit or explicit contracts are violated in the banking sector then they become emotionally exhausted and show less engagement and indulge in ineffectual silence behavior to overcome anxiety and depression. However, the following hypothesis is proposed after studying above mentioned contextual relationship.

H$_7$: Psychological contract breach moderates the relationship between workplace bullying and employee silence.

## Workplace bullying and work engagement

Work engagement is defined as psychological condition in which employees feel satisfied about their work-related activities which helps to produce effective results [31]. Engaged employees are willing to perform their work more zealously, enthusiastically, and creatively. They would try to create an environment where everyone is encouraged and motivated [57]. Moreover, research shows that there is adverse relationship between bullying and work engagement in Pakistani banking sector. To support this argument, job-demand resource theory (JD-R theory) suggests that job- demand and resources are important factors that affect directly or indirectly engaged employees [58]. However, it is argued that availability of these resources i.e., sound working conditions, managerial support in branch, positive feedback and timely promotion would help the employees to work with more dedication. Additionally, COR theory suggests that victimized employees utilize less energy for resources allocation because employees who are humiliated are ignored by their manager. They would try to sabotage the emotional resources that are important for employee's progress in any organization [59]. Therefore, based on these arguments, it is revealed that in banking sector especially where power distance culture prevails, employees are abused, miss-treated by their supervisor. It creates devastating effect on employee's inner motivation, satisfaction, and work engagement. Hence, following hypothesis is inferred from above mentioned arguments.

H$_8$: There is negative relationship between workplace bullying and work engagement.

The following theoretical model is proposed to support the above-derived hypotheses (Fig 1).

## Materials and methods

### Sample and study procedure

Cross Sectional data were collected from early-career employees having experience up to 3 years in the banking sector. The convenience sampling technique was used to get responses from seven different banks and its branches of Lahore. This technique was used because of the clear advantages of the sampling procedure and also how the participants could be reached [60]. This technique is also recommended by researchers while collecting data from the service industry [61]. According to Krejcie & Morgan (1970) table, the minimum number of sample size that was determined was 350 [62]. Overall, 384 questionnaires were distributed from

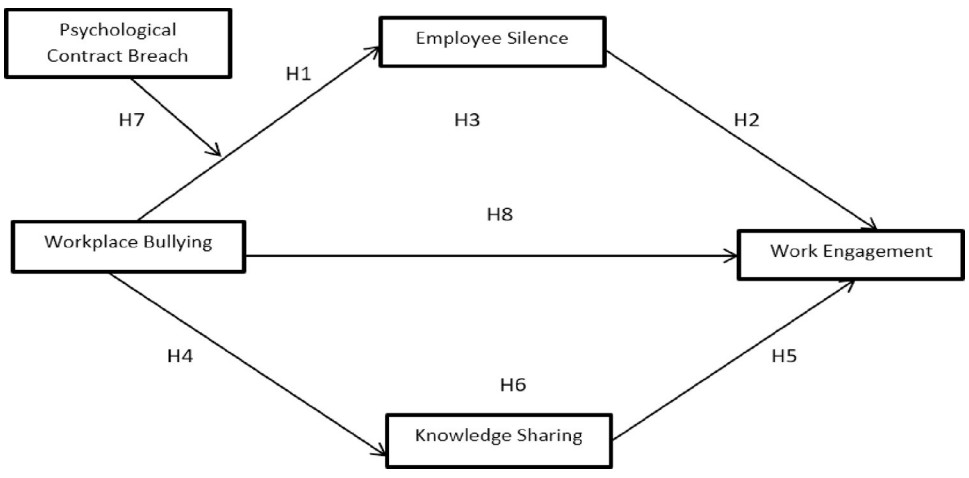

**Fig 1. Conceptual model.**

which 363 complete responses are collected via email and Google form. Online data collection was adopted because of its multiple advantages [63]. The response rate of this study was overall 95%. Within the dataset and the sample set that was available for analysis no missing value was revealed hence, there was no pattern to the missing values. Although there are some reservations attached with collection of data online. In the present research this mean was adopted since it was observed that the employees were comfortable in giving information online since they were assured of anonymity and in the forms their identity was not required. Another positive point in gathering the data online was that the topic was very sensitive i.e., bullying and thus employees would have not been comfortable in responding through a paper questionnaire. Ethical approval was obtained from the Departmental Committee of Professional Ethics FAST School of Management.

## Demographic information

For this study, demographic factors were age, gender, qualification, and length of service. However, female respondents were 59% and 41% were male participants. Education level for respondents was categorized into intermediate, bachelor, and masters. Among the respondents 65% were masters and people with bachelor qualification were 35%. Out of the total, almost 83% of the employees were in the age bracket of 21–30 years and 14% had age between 31 to 40 years and only 3% were 41 years and above. This is in line with the present research since the current research focused on early career employees. The details related to demographic factors are given in Table 1.

## Measures

In this study, all the measures of the construct were adopted from prior established studies. The detail of all measurement scales are given below.

## Workplace bullying

Workplace bullying was measured by a negative act questionnaire (NAQ- Revised) with 22 items, which was developed by Einarsen, Hoel, and Notelaers [20]. For this study, bullying behavior was measured by using two dimensions i.e., work-related and individual-related and 10 items were selected. The scale of an item was anchored from a range of 1(never) to 5

**Table 1. Demographic information.**

| Demographics | Categories | Responses | Percentage |
|---|---|---|---|
| **Gender** | Male | 150 | 41% |
| | Female | 213 | 59% |
| **Age (Years)** | 21–30 | 299 | 83% |
| | 31–40 | 51 | 14% |
| | 41 and above | 13 | 3% |
| **Educational Level** | Metric or Below | N/L | - |
| | Intermediate (FSC/FA) | N/L | - |
| | Bachelor's | 128 | 35% |
| | Masters | 235 | 65% |

(always). Some items of the negative act questionnaire (NAQ-scale) are "Being ordered to do work below your level of competence" (work-related) and "being humiliated or ridiculed in connection with your work" (person-related). The cumulative score of α coefficient was 0.97.

## Employee silence

To measure the silent behavior among employees of the banking sector a scale was adopted from the previous research conducted by Brinsfield [64]. The present study focused on two dimensions i.e. defensive silence and ineffectual silence which includes 10 items. Some of the items of these dimensions are: "I feel that sometimes speaking up is dangerous for my career (Defensive silence), I don't believe that speaking up resolves the problems (Ineffectual silence). A 5-point Likert scale was used to rate the items ranging from 1(strongly disagree) to 5 (strongly agree).

## Knowledge sharing

The items of this construct were adopted from previous research studies conducted by Van, Bart and Jan, in 2004 [65]. It contains 5 items for measurement of scale such as "I share the information with colleagues within my organization". This scale has helped to measure the knowledge-sharing behavior among early-career employees of the banking sector. A 5-point Likert scale was used to rate the items range from 1(strongly disagree) to 5(strongly agree).

## Psychological contract breach

This scale is adopted from the past research study of Robinson and Morrison [53]. For the present study, 4 items were selected for to measure psychological contract breach i.e. "I feel that my organization has breached the contract with me". For this purpose, a 5-point Likert scale was used to rate the items ranging from 1(strongly disagree) to 5(strongly agree). The coefficient of α scored for this construct was 0.90.

## Work engagement

The work engagement construct was examined by using Utrecht Work Engagement Scale (UWES-9) which included three dimensions vigor, dedication, and absorption to measure work engagement among employees [31]. However, 6 items were used for this study to measure the whole construct with its dimensions. For this purpose, a 5-point Likert scale was used to rate the items ranging from 1(strongly disagree) to 5(strongly agree). The coefficient of α scored for this construct was 0.90.

## Statistical model applied

For this study, the co-variance-based partial least square structural equation modeling (PLS-SEM) technique was applied and SmartPLS 3 software was used for data analysis. The reason behind using this tool is the effect on both types of studies i.e., exploratory and confirmatory. PLS-SEM includes two types of models for analysis: one is the measurement model and the other one is the structural equation model [66]. This technique was used since it is the most reliable method when the sample size is small and a complex model with multiple variables are involved [67]. It is helpful to evaluate the factor loadings and provides a way to decrease the parameter estimated bias.

## Results

### Measurement model

For evaluation of the measurement model PLS algorithm is applied. It helps to test the reliability, discriminant validity, and factor loadings against each item of the construct. Cronbach's Alpha, Composite Reliability and Average Variance Extract values are presented in Table 2. Reliability was tested using Cronbach's Alpha which shows the internal consistency of results, it was introduced by Lee Cronbach in 1951 [66]. It shows inter-item correlations which means that how many items of the variable are closely related to each other [67].

**Table 2. Measurement model.**

| Variables | Items | FLVs | $\alpha$ | CR | AVE |
|---|---|---|---|---|---|
| Employee Silence | ES1 | 0.849 | 0.956 | 0.963 | 0.766 |
| | ES2 | 0.885 | | | |
| | ES3 | 0.848 | | | |
| | ES4 | 0.903 | | | |
| | ES5 | 0.868 | | | |
| | ES6 | 0.875 | | | |
| | ES7 | 0.886 | | | |
| | ES8 | 0.886 | | | |
| Knowledge Sharing | KS1 | 0.924 | 0.948 | 0.960 | 0.829 |
| | KS2 | 0.916 | | | |
| | KS3 | 0.896 | | | |
| | KS4 | 0.904 | | | |
| | KS5 | 0.913 | | | |
| Psychological Contract Breach | PCB1 | 0.832 | 0.788 | 0.862 | 0.611 |
| | PCB2 | 0.771 | | | |
| | PCB3 | 0.779 | | | |
| | PCB4 | 0.741 | | | |
| Work Engagement | WE1 | 0.642 | 0.720 | 0.832 | 0.627 |
| | WE2 | 0.802 | | | |
| | WE3 | 0.909 | | | |
| Workplace Bullying | WPB1 | 0.840 | 0.853 | 0.894 | 0.629 |
| | WPB2 | 0.837 | | | |
| | WPB3 | 0.765 | | | |
| | WPB4 | 0.751 | | | |

**Note:** FLVs = Factor loading values, $\alpha$ = Cronbach's alpha, CR = Composite Reliability, AVE = Average Variance Extracted

**Table 3. Discriminant validity: Hetrotrait-Monotrait (HTMT) ratios.**

| Constructs | ES | KS | PCB | WE | WPB |
|---|---|---|---|---|---|
| ES | | | | | |
| KS | 0.143 | | | | |
| PCB | 0.419 | 0.092 | | | |
| WE | 0.233 | 0.483 | 0.138 | | |
| WPB | 0.754 | 0.133 | 0.107 | 0.321 | |

Note: ES = Employee Silence, KS = Knowledge Sharing, PCB = Psychological contract breach, WE = Work Engagement, WPB = Workplace bullying

From the empirical data, it is observed that all the values of the reliability were in the acceptable range i.e. above 0.7, this proves that all the items were reliable. The composite reliability value is also greater than 0.70 which shows that each construct items are highly internally consistent. The main aim of composite reliability is to access the reliability of each construct through the measurement of outer loading values in data [68]. According to Fornell and Larcker, (1981) criteria, average variance extracted (AVE) is mostly used to measure the amount of variance between latent constructs. However, convergent validity is accessed by using AVE having a threshold value of 0.5 [41]. Furthermore, the average variance extracted is stated as "the grand mean values of the squared loadings of the indicators associated with a particular construct (the sum of the squared loadings divided by the numbers of indicators)" [69]. For this study, all the values of AVE were above the acceptable range 0.5.

Discriminant validity depicts that how many constructs are unrelated to each other in the model i.e. constructs differ among each other. It is accessed through evaluation of cross loading values in two ways: one is Fornell-Larcker criterion and the other is Hetrotrait- Monotrait ratio (HTMT). Discriminant validity is defined as uniqueness or distinctiveness among variables whether each variable shows distinctiveness or not [69]. However, for measurement of discriminant validity self-loading values of each construct should be greater than other construct values [70]. It is measured by comparing the squared root value of average variance (AVE) by correlating with other latent constructs. Although the entire square root values of AVE are accessible in diagonal form through correlation table [71]. Therefore, for this study HTMT ratio method was applied which shows that if values of all variables are < 0.85 or up to 0.90 then they are considered in the acceptable range [72]. Table 3, shows that all the diagonal values of the construct are in acceptable range.

## Hypothesis testing and analysis

In this study, hypotheses are tested by using a bootstrapping method in which 5000 subsamples are evaluated to get the significance level [69]. In which p-value, t- value, beta value and standard deviation were calculated.

Results of Table 4 shows the direct relation between workplace bullying and work engagement is statistically proved negative and significant (β = -0.123, p-value = 0.000). Therefore, it is concluded that $H_8$ is accepted. It is proved from the results that the direct effect between workplace bullying and employee silence is positive and significant (β = 0.644, p = 0.000). Hence, $H_1$ is supporting the current model and accepted. Subsequently negative association between employee silence and work engagement is statistically significant (β = -0.268, p = 0.00). Therefore, the $H_2$ hypothesis is accepted and supported the study. Furthermore, workplace bullying and knowledge sharing is negatively related to each other (β = -0.100, p = 0.016), it is proved that $H_4$ is accepted and supported. Likewise, the direct relation between knowledge sharing and work engagement is statistically significant and positive (β = 0.483,

**Table 4. The direct and indirect effect.**

| Construct | Beta-value $\beta$ | Standard Deviation (SD) | T- Statistics (t-value) | P- Statistics (p-values) | Decision |
|---|---|---|---|---|---|
| **Direct Effects** | | | | | |
| WPB -> ES | 0.644 | 0.035 | 18.822 | 0.000 | Supported |
| ES -> WE | -0.268 | 0.057 | 4.690 | 0.000 | Supported |
| WPB -> KS | -0.100 | 0.047 | 2.142 | 0.016 | Supported |
| KS -> WE | 0.483 | 0.037 | 13.025 | 0.000 | Supported |
| PCB -> ES | 0.287 | 0.045 | 6.386 | 0.000 | Supported |
| WPB -> WE | -0.123 | 0.057 | 2.181 | 0.000 | Supported |
| **Indirect Effects** | | | | | |
| WPB -> ES ->WE | -0.040 | 0.039 | 4.543 | 0.000 | Supported |
| WPB -> KS ->WE | -0.048 | 0.023 | 2.064 | 0.020 | Supported |
| Moderating Effect 1> ES | 0.235 | 0.043 | 5.496 | 0.000 | Supported |

p = 0.000). Hence, from the empirical data it can be observed that the H$_5$ is accepted. Moreover, the indirect effect is calculated through bootstrapping procedure in Smart-PLS. The indirect relationship of employee silence between workplace bullying and work engagement is significant ($\beta$ = -0.178, p-value = 0.00). Likewise, the indirect relationship of knowledge sharing among workplace bullying and work engagement is also significant ($\beta$ = -0.048, p-value = 0.020). Hence, it is concluded that all the direct and indirect paths are significant and partial mediation is justified. Subsequently, in moderation analysis, the moderating variable psychological contract breach is highly significant ($\beta$ = 0.235, p-value = 0.000). However, the results from the empirical data show that psychological contract breach positively moderates the mediating relationship of workplace bullying and employee silence.

## Discussion

Employees are the intellectual capital for an organization and for this many organizations are diverting their attention to improve the mental wellbeing of the employees. Mental wellbeing is an important component to keep the employees engaged in their work and encourage them to share knowledge among each other. This concept regarding workplace environment and mental health of employees have gained a lot of attention of the researchers as well. The main purpose of this study was to measure the effect of workplace bullying on work engagement among early-career banking employees through employee silence and knowledge sharing. Moreover, the study explains how PCB moderates the relationship between employee silence and bullying. In the present study, the data were collected from the early career employees of different banks and its branches of Lahore, Pakistan. The results revealed that employee silence and knowledge sharing partially mediated the bullying-engagement relationship. Thus, it was concluded that workplace bullying has both direct and indirect effect on work engagement. It was evident when mediation was tested that the silent behavior of employees leads towards less engaged behavior of employees due to dissatisfaction and adverse workplace environment. These results were also in accordance with the past studies that were conducted in different organizations of India and universities of Pakistan [53, 71]. Literature states that silent behavior of staff members is a passive coping strategy for workplace bullying. Limiting the extent to share the knowledge and other important resources and information is also a defensive way to overcome the future loss of internal resources due to bullying [60]. Further, the effect of PCB as a moderator among workplace bullying and employee silence is consistent with the results from prior literature conducted in the software houses of Pakistan [52]. It is clear from the results that the moderation effect of PCB positively strengthens the bulling-silence

relationship. It is suggested from the results that when early-career employees are bullied in banks and their promises are not fulfilled then they ultimately adopt the strategy to stay silent for future security. Employees who are victims of workplace bullying tend to hide the knowledge that they have and not share it with other employees and hence are not engaged in their work as well. Past research also supports the argument that when employees' psychological promises (i.e., recognition, work-related admiration, bonuses, and effective feedback) are fulfilled, they are less victimized with bullied behavior [25]. Moreover, the results of this study are complemented by social contract theory and COR theory [11, 13] which states that early career banking employees try to conserve their inner resources and adopt silent behavior, when their promises are not fulfilled.

Employee engagement can be considered as a potential self-protective behavior that buffers against bullying because engaged employees are more likely to take proactive steps to address abusive behavior and prevent it from continuing. Engaged employees are more likely to feel invested in their work and their workplace, and as a result, they may be more likely to speak up and report incidents of bullying, or to actively work to prevent bullying from occurring in the first place. On the other hand, employee silence can be viewed as allowing bullies to get away with their behavior, which can lead to a vicious cycle of abuse. When employees remain silent, bullies are able to continue their behavior without consequences, and this can create a toxic workplace culture where bullying is normalized and allowed to persist. Additionally, when employees remain silent, they may also be viewed as passive accomplices to the bullying behavior, which can further contribute to the cycle of abuse. Therefore, by framing employee engagement as a self-protective behavior, organizations can help to encourage employees to take an active role in preventing bullying and promoting a positive workplace culture. Conversely, by highlighting the negative consequences of employee silence, organizations can help to discourage employees from remaining passive and instead encourage them to take action to address bullying and create a safer and more positive workplace environment.

The results of the present study highlights that negative workplace environment in the form of bullying and mistreatment negatively impacts the early-career employee's hence they exhibit less motivation and satisfaction to perform their banking tasks. It is also concluded that if the managers are not supportive in banks and do not provide their employees with an environment of security, then the employees would not share any effective information and knowledge with colleagues [8]. However, it is recommended that the banks should have a monitored environment and give confidence to the employees to discuss their issues to overcome the adverse consequences of less-engaged behavior among early-career banking employees [66]. Furthermore, the culture of negative workplace behavior hinders the performance, innovation, and creativity among the early-career banking employees. Prior research shows that the silent behavior of banking employees in response to workplace bullying is a strategy to overcome the issues at the employee level [69]. It can severely damage the work engagement, performance, and productivity of the organization. The following are the managerial implications to overcome this dilemma in the future occurrence. First, a manager should ensure an environment where workplace bullying is monitored at each hierarchical level within the bank. The Human Resource Department should play a vital role in making and implementing policies which are helpful in this regard. Multiple negative outcomes may occur due to an environment where mistreatment occurs. Employee silence is a consequence of bullying among the early-career victimized employees, where they may also opt to limit their knowledge sharing to overcome the fear. Second, at the organizational and departmental level, a trustworthy and supportive relationship should be maintained among employees and manager. This relationship would help to improve the confidence among early-career banking employees to report any misconduct. Third, proper training and awareness sessions, informal gatherings and social meetings

should be conducted especially for early-career banking employees. This may help to build the confidence and make employees internally satisfied to do their work with more dedication. Forth, the management should also develop and ensure the implementation of awareness programs where the employees are given training regarding their rights and file reservations if they have any.

## Limitations and future directions

This section presents some of the limitations of the present study and also some future directions that are derived. First, the data was collected from the employees of the banking industry hence, generalizing it to other industries may not be possible. Second, knowledge sharing and employee silence is taken as mediators in this study, it is suggested for future research to use variables like job stress, future career anxiety and social support as mediators to observe different perspectives of the impact of workplace bullying. Third, the present research only considers cross-sectional design. Further, the same model may be applied on longitudinal studies to open new horizons of research in the future. Forth, the data was collected only from the early career employees to understand and comprehend the impact of workplace bullying on them specially since it is thought that employees in their mid-career are better able to handle the situation. Fifth, convenience sampling was used to select the early career employee of the banking sector hence, the results cannot be generalized to all banks of the regions of Pakistan. For future probability sampling should be expanded to other provinces as well. Despite these limitations, this study contributes to the literature in multiple ways and among them the most important is that it is specific to early career employees. It is observed that employees who have experience are able to cope up with bullying but early career employees are the one who are the most effected with such complications. We believe that this study will aid the managers in setting up guidelines, preventive measure, and support human resource managers in making policies that consider these factors important while developing strategies. To accomplish this, it is important to counsel the early career employees and make sure that there is no communication gap between the authorities and the employees.

## Conclusions

Organizations that want to survive the continuous changing and dynamic environment, needs employees to share resources and knowledge among each other to achieve competitive edge. This cannot be done when employees are not comfortable with the culture and organizational setting where workplace bullying prevails. Workplace bullying is one of the major concerns for employers as well as employees. Previous research proves that workplace bullying has negative impact on work engagement however, this research highlights the importance of employee silence and knowledge sharing having a greater impact on work engagement. This study also highlights that employees choose to have a silent behavior if they encounter workplace bullying and consider that there is a psychological contract breach. Current research intensifies that the negative workplace environment in the form of bullying and mistreatment among early-career employee's exhibit less motivation and work-engagement to perform their daily activities. Meanwhile, the findings of result shows that the employees silence and knowledge sharing are directly related to the workplace bullying. It negatively relates with work engagement to hinder their performance. However, the results suggest that early-career employees in banks exhibit silent behavior due to victimization and bullying activities. Therefore, managers in banks should devise strategies to promote anti-bullying culture and supportive climate, where early-career employees can raise their issues without any organizational pressure. An environment of concern and openness should be encouraged and proper policies should be devised by the

employers to discourage workplace bullying and handle such situations. Proper channel of communication must be guided to the early employees so they can discuss their issues if they have any.

## Acknowledgments

We would like to thank the employees in the bank for giving us time during response collection and also M. Arif Khan Niazi for coordinating while collecting all the data.

## Author Contributions

**Conceptualization:** Amna Niazi.

**Data curation:** Iqra Javed.

**Formal analysis:** Amna Niazi.

**Investigation:** Iqra Javed.

**Project administration:** Mujahid Hussain.

**Supervision:** Sadia Nawaz, Mujahid Hussain.

**Validation:** Sadia Nawaz, Muhammad Ali.

**Writing – original draft:** Muhammad Ali.

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
