## [Decision Letter · Decision Letter 0]

21 Sep 2022

PONE-D-22-23996Impact of Workplace Bullying on Work Engagement among Early Career EmployeesPLOS ONE

Dear Dr. Niazi,

Thank you for submitting your manuscript to PLOS ONE. After careful consideration, we feel that it has merit but does not fully meet PLOS ONE’s publication criteria as it currently stands. Therefore, we invite you to submit a revised version of the manuscript that addresses the points raised during the review process.

Specific comments appears at the end of this letter. Please go through them carefully and prepare your revision accordingly.

We look forward to receiving your revised manuscript.

Kind regards,

Rajneesh Choubisa, Ph.D

Academic Editor

PLOS ONE

Journal Requirements:

3. Please provide additional details regarding ethical approval in the body of your manuscript. In the Methods section, please ensure that you have specified the name of the IRB/ethics committee that approved your study.

Additional Academic Editor Comment:

Dear Author/s

This article captures the essence of workplace bullying in the context of Pakistan and follows a decent scientific approach. However, the reviewers highlight that the arguments in favor of the sample and the choice of the sector is carried out weakly and require improvement. Also, the author/s should come up with their operational definition of workplace bullying because it might have different connotations among different sectors. All reviewers, including myself, feel that adequate justification and rationale should be added for the sample under study. Besides, all reviewers feel that the discussion is lacking an overall explanation and support for the current findings and need to be rewritten wherein the authors can provide more support to their model with the help of relevant literature. A revised version that addresses the concerns might be a good fit for the journal.

Best Wishes,

Academic Editor

Reviewers' comments:

Reviewer's Responses to Questions

**Comments to the Author**

1. Is the manuscript technically sound, and do the data support the conclusions?

Reviewer #1: Yes

Reviewer #2: Yes

Reviewer #3: Yes

Reviewer #4: Yes

Reviewer #5: Yes

2. Has the statistical analysis been performed appropriately and rigorously? 

Reviewer #1: Yes

Reviewer #2: Yes

Reviewer #3: Yes

Reviewer #4: Yes

Reviewer #5: Yes

3. Have the authors made all data underlying the findings in their manuscript fully available?

Reviewer #1: No

Reviewer #2: Yes

Reviewer #3: No

Reviewer #4: No

Reviewer #5: Yes

4. Is the manuscript presented in an intelligible fashion and written in standard English?

Reviewer #1: Yes

Reviewer #2: Yes

Reviewer #3: Yes

Reviewer #4: No

Reviewer #5: Yes

5. Review Comments to the Author

Reviewer #1: 1. The introduction and literature review is lengthy and difficult to follow, especially the concepts of workplace bulling can be concisely written. The reference in entire article have to be modified little bit to suit the reference style adopted by the journal. It seems the article was originally written in other form of referencing and have been converted to the current form. Hence fine tuning is needed.

3. The model used by authors provide minimal innovation and has been reported by several authors in different context. The authors also do not provide the demographic details of the sample by which one can learn about the situation relatively better way.

4. The researchers used a very haphazard method of convenient sampling method which undermine the strength of the study. Now a days many researchers just conducting some online survey with members of near and dear circle and reporting to the reputed journals with little justifications. Hence the authors are advised to provide irrefutable proof in the methodology to justify the sample.

5. The discussion is very scant and need to support studies in sector wise manner because there may be the bulling effect is different from one sector to other.

6. The authors have to reduce the writing portion before deriving the hypothesis.

Reviewer #2: I express my gratitude for providing me with the opportunity to review the manuscript. Reviewing the paper was a great learning experience for me. In this manuscript the authors depicted an interesting study on workplace bullying in the context of Pakistan.

Here are some of my observations and suggestions for further improvement.

Introduction and Literature review

1. Full form of COR [conservation of resource theory] [page 4, line 75] should be mentioned first.

2. Line 87: Different in-text citation was used. Also the sentence need clarity.

3. Lines 96- 104_contribution: the authors need to elaborate this section by adding the rationale of the study. Lack of exploration in Pakistan is an important point. However, the argument can be improved by adding more about some points, such as, why this study is important for the population in question; are there some Government related initiatives of workplace bullying; the cultural aspects related to different societal categories etc.; how these grievances are reported and whether there is a formal committee to look after these issues or it is ignored etc.

4. Line 107: Since the authors have mentioned “hypothesis” first time in this section, it is advisable to mention hypothesis in previous section where the research aim (line 91) was mentioned.

5. The literature review is extensive and relevant. The hypotheses are clearly depicted. However, my suggestion would be to make this section more concise. A specific section on cultural aspect (e.g. work place culture, diversity, gender representation, self-construal etc.) relevant in the context of Pakistan can be also added. Right now this section is lucid in terms of language.

Materials and methods

6. Lines 319, 328, and 336: the author’s name should be mentioned.

7. The authors can elaborate on ethical issues concerning online data collection.

8. Authors can mention about the time line of data collections, data organisation (handling missing data etc. ), and ethics board approval in the main document.

Discussion / Conclusion

9. The authors can elaborate on the implications of the study in terms of work related policy, development of awareness program etc.

Reviewer #3: There are a few issues in the overall flow of the manuscript. The introduction needs to be formatted ensuring the explanations of all key variables and concepts are there. Ex: Some abbreviations like OCR have been used in the introductory lines but not explained in the beginning but only towards a later point in the manuscript. APA 7 has not been followed thoroughly in the manuscript and needs to be revisited for smaller typos and formatting issues. The model testing is robust and the results section seems to do justice to the research questions. However, the discussion is lacking an overall explanation and support to the current findings, the authors can provide more support to their model by relevant literature. The manuscript has a scope of improvement in the Intro and Discussion sections. The authors are also encouraged to give the strengths and limitations of their work along with the future directions to ensure scientific progression in this area of research.

Reviewer #4: 1. There were grammatical errors in the paper which needs to be corrected.

2. A systematic structure was missing in the article (Design, Objectives, Limitations, Future Implications).

3. Work Engagement as a variable hasn't been elaborated much. What are the components for it?

4. The author hasn't provided much literary association between workplace bullying and work engagement.

5. The paper is vague in it's structure. More reasoning shall be provided for the choice of the variables.

6. Discussion section didn't justify the results mentioned. Again a systematic form is required and more research supported can be added.

7. Re-write the Methodology part and make it concise and explicit.

8. Would like to know - why did you specifically choose this study and what makes it unique? How is it different from other studies out there?

9. Have you applied anywhere else other than PLOS ONE?

Reviewer #5: The manuscript titled "Impact of Workplace Bullying on Work Engagement among Early Career Employees" has been conceptualized well. The theoretical background i.e. COR theory in relation to the proposed objectives of the study have been explained clearly by the authors. The topic of research is very relevant.

However, minor revision is required to elevate the quality of the manuscript:

1. The manuscript requires more detailed description of literature gap.

2. When describing limitations of the study, more emphasis could be made on how to overcome and/or tackle the issue of workplace bullying.

3. Mention how has the sample size been estimated? Also there seems to be ambiguity in the age range of the sample. The title mentions "early career employees" but the demographic age related demographic details seem contradictory.

4. The discussion and conclusion need more elaboration.

5. Revisions are required to correct some phrasing and grammatical errors across the manuscript. Some repetition has been observed in the introduction. Copyediting is suggested.

6. On page no 5, the structure of the paper need not be mentioned.

6. PLOS authors have the option to publish the peer review history of their article (what does this mean?). If published, this will include your full peer review and any attached files.

Reviewer #1: **Yes: **Ranjit Kumar Dehury

Reviewer #2: No

Reviewer #3: **Yes: **Dr. Garima Rajan

Reviewer #4: No

Reviewer #5: No

---

## [Author Response · Author response to Decision Letter 0]

10 Nov 2022

Reviewer 1

Comment 1

The introduction and literature review is lengthy and difficult to follow, especially the concepts of workplace bulling can be concisely written. The reference in entire article have to be modified little bit to suit the reference style adopted by the journal. It seems the article was originally written in other form of referencing and have been converted to the current form. Hence fine tuning is needed.

Response:

The literature review has been modified. References are converted through Mendeley software and hence rechecked and fine tuned according to the journal requirements. 

Comment 2

The model used by authors provide minimal innovation and has been reported by several authors in different context. The authors also do not provide the demographic details of the sample by which one can learn about the situation relatively better way.

Response:

Relevant Literature has been added in the introduction to prove the uniqueness of the model. The demographic information regarding the sample is detailed in table 1.

Comment 3

The researchers used a very haphazard method of convenient sampling method which undermine the strength of the study. Now a days many researchers just conducting some online survey with members of near and dear circle and reporting to the reputed journals with little justifications. Hence the authors are advised to provide irrefutable proof in the methodology to justify the sample.

Response:

As per recommendation, proof has been added in the methodology section detailing why the convenience sampling was used. 

Comment 4

The discussion is very scant and need to support studies in sector wise manner because there may be the bulling effect is different from one sector to other.

Response:

Discussion has been rewritten for clarity. Industries are now mentioned in the discussion section of the article. 

Comment 5

The authors have to reduce the writing portion before deriving the hypothesis.

Response:

The writing part has been reduced. The structure of the paper mentioned in the introduction was also removed.

Reviewer #2

Comment 1

Full form of COR [conservation of resource theory] [page 4, line 75] should be mentioned first.

Response:

The mistake has been rectified. 

Comment 2

Line 87: Different in-text citation was used. Also the sentence needs clarity.

Response:

The reference has been changed to the journal guidelines and the sentences are changed to bring more clarity. 

Comment 3

Lines 96- 104_contribution: the authors need to elaborate this section by adding the rationale of the study. Lack of exploration in Pakistan is an important point. However, the argument can be improved by adding more about some points, such as, why this study is important for the population in question; are there some Government related initiatives of workplace bullying; the cultural aspects related to different societal categories etc.; how these grievances are reported and whether there is a formal committee to look after these issues or it is ignored etc.

Response:

Thank you for the detailed feedback. The recommendations were incorporated, and rationale of the study was also elaborated. 

Comment 4

Line 107: Since the authors have mentioned “hypothesis” first time in this section, it is advisable to mention hypothesis in previous section where the research aim (line 91) was mentioned.

Response:

It is now mentioned in the research aim regarding the hypotheses development. 

Comment 5

The literature review is extensive and relevant. The hypotheses are clearly depicted. However, my suggestion would be to make this section more concise. A specific section on cultural aspect (e.g. work place culture, diversity, gender representation, self-construal etc.) relevant in the context of Pakistan can be also added. Right now this section is lucid in terms of language.

Response:

The mentioned feedback has been added at multiple places in the article. 

Comment 6

Lines 319, 328, and 336: the author’s name should be mentioned.

Response:

The authors names are now mentioned in the recommended lines.

Comment 7

The authors can elaborate on ethical issues concerning online data collection.

Response:

It is now mentioned in detail in the sample and study procedure section in the paper. 

Comment 8

Authors can mention about the time line of data collections, data organization (handling missing data etc. ), and ethics board approval in the main document.

Response:

All the mentioned points are now mentioned in the sample and study section including the ethics board approval. 

Comment 9

The authors can elaborate on the implications of the study in terms of work-related policy, development of awareness program etc.

Response:

Policies by the organizations were already mentioned and the awareness programs are now mentioned in the implications. 

Reviewer #3: 

Comment 1

There are a few issues in the overall flow of the manuscript. The introduction needs to be formatted ensuring the explanations of all key variables and concepts are there. Ex: Some abbreviations like OCR have been used in the introductory lines but not explained in the beginning but only towards a later point in the manuscript. 

Response:

Thank you for pointing this out. The mistakes have been rectified. 

Comment 2

APA 7 has not been followed thoroughly in the manuscript and needs to be revisited for smaller typos and formatting issues. 

Response:

The required format for reference for this journal is Vancouver and hence referencing has been changed to the required format through Mendeley software. 

Comment 3

The model testing is robust and the results section seems to do justice to the research questions. However, the discussion is lacking an overall explanation and support to the current findings, the authors can provide more support to their model by relevant literature. 

Response:

The discussion section has be re-written to incorporate the mentioned feedback.

Comment 4

The manuscript has a scope of improvement in the Intro and Discussion sections. The authors are also encouraged to give the strengths and limitations of their work along with the future directions to ensure scientific progression in this area of research.

Response:

Introduction and discussion sections have been improved. Strengths have been mentioned while discussing the objective of the study and the limitations and future directions are mentioned in the end of the article. 

Reviewer #4: 

Comment 1

There were grammatical errors in the paper which needs to be corrected.

Response:

The paper has been checked multiple times for correction of grammatical errors. 

Comment 2

A systematic structure was missing in the article (Design, Objectives, Limitations, Future Implications).

Response:

The structure has been fixed. 

Comment 3

Work Engagement as a variable hasn't been elaborated much. What are the components for it?

Response:

The explanation or work engagement and its components have been added to the introduction. 

Comment 4

The author hasn't provided much literary association between workplace bullying and work engagement.

Response:

The association between workplace bullying and work engagement has been added in the literature. 

Comment 5

The paper is vague in it's structure. More reasoning shall be provided for the choice of the variables.

Response:

Amendments have been made to improve the structure of the paper and reasoning has been added explaining why these variables were used. 

Comment 6

Discussion section didn't justify the results mentioned. Again, a systematic form is required, and more research supported can be added.

Response:

To improve the discussion section it was rewritten to improve the systematic form and also research form literature was added to support the discussion. 

Comment 7

Re-write the Methodology part and make it concise and explicit.

Response:

The methodology part was re-written to clear the ambiguity and details were also added. 

Comment 8

Would like to know - why did you specifically choose this study and what makes it unique? How is it different from other studies out there?

Response:

This is mentioned in the paper however, the study incorporates psychological contract breach as an important and significant variable and according to authors best knowledge there is no study that has focused on early career employees in Pakistan. This is an important point since experienced employees can handle disturbing situations however, early career employees are naïve and don’t know how to cope up with certain situations. This was the main reason why this study focused on the variables that were selected.

Comment 9

Have you applied anywhere else other than PLOS ONE?

Response:

Currently, the article is not sent to any other journal other than PLOS one. The present article was sent to a journal before PLOS ONE but the scope of the journal did not match with the current article. 

Reviewer #5: 

Comment 1

The manuscript requires more detailed description of literature gap.

Response:

The description on literature gap has been added. 

Comment 2

When describing limitations of the study, more emphasis could be made on how to overcome and/or tackle the issue of workplace bullying.

Response:

Limitations have been rewritten with emphasis on how to overcome bullying at workplace. 

Comment 3

Mention how has the sample size been estimated? Also there seems to be ambiguity in the age range of the sample. The title mentions "early career employees" but the demographic age related demographic details seem contradictory.

Response:

The estimation of the sample size has been determined. The age brackets are explained in the demographic section. 

Comment 4

The discussion and conclusion need more elaboration.

Response:

The discussion and conclusion is now elaborated. 

Comment 5

Revisions are required to correct some phrasing and grammatical errors across the manuscript. Some repetition has been observed in the introduction. Copyediting is suggested.

Response:

The manuscript has been checked multiple times for correction of grammatical as well mistakes related to repetition. 

Comment 6

On page no 5, the structure of the paper need not be mentioned

Response:

The structure that was mentioned on page 5 has been removed.

---

## [Decision Letter · Decision Letter 1]

30 Jan 2023

PONE-D-22-23996R1Impact of Workplace Bullying on Work Engagement among Early Career EmployeesPLOS ONE

Dear Dr. Niazi,

Thank you for submitting your manuscript to PLOS ONE. After careful consideration, we feel that it has merit but does not fully meet PLOS ONE’s publication criteria as it currently stands. Therefore, we invite you to submit a revised version of the manuscript that addresses the points raised during the review process.

My suggestions and observations appear at the end of this letter.

 Please submit your revised manuscript by Mar 16 2023 11:59PM. If you will need more time than this to complete your revisions, please reply to this message or contact the journal office at plosone@plos.org. Please include the following items when submitting your revised manuscript:A rebuttal letter that responds to each point raised by the academic editor and reviewer(s). You should upload this letter as a separate file labeled 'Response to Reviewers'.A marked-up copy of your manuscript that highlights changes made to the original version. You should upload this as a separate file labeled 'Revised Manuscript with Track Changes'.An unmarked version of your revised paper without tracked changes. You should upload this as a separate file labeled 'Manuscript'.

We look forward to receiving your revised manuscript.

Kind regards,

Rajneesh Choubisa, Ph.D

Academic Editor

PLOS ONE

Additional Editor Comments (if provided):

Dear Authors,

The revised manuscript looks promising but it still requires more streamlining on the grounds that can warrant its publication. The authors suggest that it is interesting to find out how bullied employees are going to behave when it comes to knowledge sharing especially when already plenty of research on negative effects of bullying is available. What is missing in the manuscript is why the specific effects authors have investigated are interesting and relevant in view of a host of existing literature on bullying effects. More justification in terms of the characteristics of the sample (sector, tenure, and country) that makes it a unique and meaningful contribution should be added for further improvement.

Kindly rework on the manuscript to further improve the manuscript and make it soulful.

Best Wishes,

Academic Editor

Reviewers' comments:

Reviewer's Responses to Questions

**Comments to the Author**

1. If the authors have adequately addressed your comments raised in a previous round of review and you feel that this manuscript is now acceptable for publication, you may indicate that here to bypass the “Comments to the Author” section, enter your conflict of interest statement in the “Confidential to Editor” section, and submit your "Accept" recommendation.

Reviewer #6: All comments have been addressed

2. Is the manuscript technically sound, and do the data support the conclusions?

Reviewer #6: Partly

3. Has the statistical analysis been performed appropriately and rigorously? 

Reviewer #6: Yes

4. Have the authors made all data underlying the findings in their manuscript fully available?

Reviewer #6: No

5. Is the manuscript presented in an intelligible fashion and written in standard English?

Reviewer #6: Yes

6. Review Comments to the Author

Reviewer #6: In the present manuscript, the authors examine the relationship between experiences of workplace bullying and employee work engagement, using a cross-sectional data set. Although I very much appreciate their efforts, I do have several concerns which will be discussed below.

INTRODUCTION AND THEORETICAL CONTRIBUTION

In their introduction, the authors elaborate at length on how bullying has negative impact on individuals and organizations, highlighting the practical relevance of their research area. However, a compelling theoretical contribution of the present research, embedded in a convincing story, is largely missing.

In my view, it is a key requirement for the authors to identify and describe a theoretical contribution that warrants publication. So far, the authors pointed out that “It is interesting to find out how bullied employees are going to behave when it comes to knowledge sharing within the organization” (81-83), while conceding that there is already plenty of research on negative effects of bullying. The authors largely miss to elaborate, why the specific effects they investigated are interesting and relevant in view of a host of existing literature on bullying effects. Moreover, I am not convinced that the characteristics of the sample (sector, tenure, and country) constitute a meaningful contribution, as the authors do not explain why (and test if) these characteristics might influence the investigated effects.

Notably, I do see several potential contributions in the paper that could be further developed by the authors.

First, the authors could elaborate more on their DVs and why they are novel in bullying research and/or contribute to solve existing theoretical or practical puzzles. For instance, the authors could consider framing Employee Engagement as a potential “self-protective” behavior buffering against (future) bullying, whereas Employee Silence allows bullies to get away with it and thus might even exacerbate abusive behavior in a vicious cycle.

Second, the authors could elaborate more on the moderator and how it contributes to explain and reconcile empirical inconsistencies in the field.

Third, the authors could elaborate on the context (Pakistani banking sector), going beyond claiming that this context has never been looked at, but arguing why comparability with previously studied contexts is severely limited and, why the context of the present study is relevant to expanding our overall knowledge in this field.

THEORETICAL BACKGROUND AND THEORETICAL MODEL

The authors describe both the independent and the dependent variables in a comprehensive manner. Moreover, drawing on relevant and established theory (COR), the authors introduce a compelling psychological mechanism explaining their findings.

However, I would like to point to logical flaws in the theoretical model.

First, I have some doubts about described causal links between mediators (i.e., employee silence, knowledge sharing) and the DV (i.e., work engagement). While all three variables are certainly highly correlated, I am not convinced about the described direction of the effect (i.e., that behaviors like silence and knowledge sharing lead to a more perception-based variable like work engagement). Second, I am not convinced of the interaction effect of Breaching of a Psychological Contract and workplace bullying on employee engagement, given that the authors miss to explain the hypothesized interaction between PCB and bullying, including the simple effects of bullying if PCB is high vs. low.

To address these concerns, I encourage the authors to elaborate more on the causal links and the interaction effect they seek to establish.

METHODS AND RESULTS

The sample is described with due thoroughness.

As conceded by the authors, a major limitation is the design of the present study. Therefore, the authors cannot draw conclusions about the causality of the relationships described in the theoretical model, limiting the possibility of making a true contribution.

In my view, the present manuscript could benefit greatly from adding another study. For instance, a study with multiple points of measurement (e.g., event sampling) would help to gain insight into the direction of the effects (e.g., one could hypothesize that bullying leads to Employee Silence which in turn exacerbates bullying). Moreover, the authors may overcome the common-source bias of the present study in another study.

STYLE AND WRITING

Improvements have been made in terms of conciseness and writing.

7. PLOS authors have the option to publish the peer review history of their article (what does this mean?). If published, this will include your full peer review and any attached files.

Reviewer #6: No

---

## [Author Response · Author response to Decision Letter 1]

8 Mar 2023

Thank you for the opportunity of improving the article. The response to the reviewers comment's have been uploaded in a separate file. 

Regards,

---

## [Editor Report · Decision Letter 2]

20 Apr 2023

Impact of Workplace Bullying on Work Engagement among Early Career Employees

PONE-D-22-23996R2

Dear Dr. Niazi,

We’re pleased to inform you that your manuscript has been judged scientifically suitable for publication and will be formally accepted for publication once it meets all outstanding technical requirements.

Kind regards,

Faisal Shafique Butt

Academic Editor

PLOS ONE
---

## [Editor Report · Acceptance letter]

22 May 2023

PONE-D-22-23996R2 

Impact of workplace bullying on work engagement among early career employees 

Dear Dr. Niazi:

I'm pleased to inform you that your manuscript has been deemed suitable for publication in PLOS ONE. Congratulations! Your manuscript is now with our production department. 

Kind regards, 

on behalf of

Dr. Faisal Shafique Butt 

Academic Editor

PLOS ONE